# Highly Pathogenic Avian Influenza H5N1 Virus Infections in Wild Red Foxes (*Vulpes vulpes*) Show Neurotropism and Adaptive Virus Mutations

Luca Bordes,[a] Sandra Vreman,[a] Rene Heutink,[a] Marit Roose,[a] Sandra Venema,[a] Sylvia B. E. Pritz-Verschuren,[a] Jolianne M. Rijks,[b] José L. Gonzales,[a] Evelien A. Germeraad,[a] Marc Engelsma,[a] Nancy Beerens[a]

[a]Wageningen Bioveterinary Research, Lelystad, the Netherlands
[b]Dutch Wildlife Health Centre, Utrecht University, Utrecht, the Netherlands

**ABSTRACT** During the 2020 to 2022 epizootic of highly pathogenic avian influenza virus (HPAI), several infections of mammalian species were reported in Europe. In the Netherlands, HPAI H5N1 virus infections were detected in three wild red foxes (*Vulpes vulpes*) that were submitted with neurological symptoms between December of 2021 and February of 2022. A histopathological analysis demonstrated that the virus was mainly present in the brain, with limited or no detection in the respiratory tract or other organs. Limited or no virus shedding was observed in throat and rectal swabs. A phylogenetic analysis showed that the three fox viruses were not closely related, but they were related to HPAI H5N1 clade 2.3.4.4b viruses that are found in wild birds. This suggests that the virus was not transmitted between the foxes. A genetic analysis demonstrated the presence of the mammalian adaptation E627K in the polymerase basic two (PB2) protein of the two fox viruses. In both foxes, the avian (PB2-627E) and the mammalian (PB2-627K) variants were present as a mixture in the virus population, which suggests that the mutation emerged in these specific animals. The two variant viruses were isolated, and virus replication and passaging experiments were performed. These experiments showed that the mutation PB2-627K increases the replication of the virus in mammalian cell lines, compared to the chicken cell line, and at the lower temperatures of the mammalian upper respiratory tract. This study showed that the HPAI H5N1 virus is capable of adaptation to mammals; however, more adaptive mutations are required to allow for efficient transmission between mammals. Therefore, surveillance in mammals should be expanded to closely monitor the emergence of zoonotic mutations for pandemic preparedness.

**IMPORTANCE** Highly pathogenic avian influenza (HPAI) viruses caused high mortality among wild birds from 2021 to 2022 in the Netherlands. Recently, three wild foxes were found to be infected with HPAI H5N1 viruses, likely due to the foxes feeding on infected birds. Although HPAI is a respiratory virus, in these foxes, the viruses were mostly detected in the brain. Two viruses isolated from the foxes contained a mutation that is associated with adaptation to mammals. We show that the mutant virus replicates better in mammalian cells than in avian cells and at the lower body temperature of mammals. More mutations are required before viruses can transmit between mammals or can be transmitted to humans. However, infections in mammalian species should be closely monitored to swiftly detect mutations that may increase the zoonotic potential of HPAI H5N1 viruses, as these may threaten public health.

**KEYWORDS** fox, HPAI H5N1, neurotropism, virology, zoonotic infections

Address correspondence to Nancy Beerens, nancy.beerens@wur.nl.

The authors declare no conflict of interest.

Since the introduction of highly pathogenic avian influenza virus (HPAI) H5 clade 2.3.4.4b in 2016, this virus clade has caused repeated outbreaks in wild birds and poultry in Europe. Whereas low pathogenic avian influenza viruses replicate mostly in the digestive and respiratory epithelium with often mild disease in poultry and wild

birds, these HPAI viruses can cause severe systemic disease, including viremia that leads to the diffuse infection of several internal organs. Infections with HPAI viruses of subtypes H5 and H7 are a notifiable disease in poultry, and after the detection of this virus, a poultry flock is generally culled to prevent further spread. The 2020 to 2021 epizootic was a devastating outbreak for poultry and was followed by the 2021 to 2022 epizootic, which has become the largest HPAI outbreak in terms of the number of culled animals that has ever occurred in Europe. In addition, high mortalities in an increasing number of species of wild birds were observed during both epizootics. Sporadically, HPAI H5 clade 2.3.4.4.b infections of free-living, wild carnivore species were reported in addition to the infections in wild birds and poultry. In late 2020, a disease and mortality event involving four juvenile common seals (*Phoca vitulina*), one juvenile gray seal (*Halichoerus grypus*), and a red fox (*Vulpes vulpes*) at a wildlife rehabilitation center in the United Kingdom was reported to be associated with HPAI H5N8 infections (1). In May 2021, HPAI H5N1 infections were reported in two cubs of red foxes in the Netherlands (2). In August 2021, HPAI H5N8 infections were detected in three adult harbor seals (*Phoca vitulina*) found at the German North Sea coast (3). Interestingly, neurological signs were reported for several of these mammals, and virus was detected in brain tissue. For mammals, the most probable route of infection is via the ingestion of contaminated water, feces, or infected bird carcasses. Foxes were infected experimentally via consumption of chicken carcasses infected with clade 2.2 HPAI H5N1, which underlines the possibility to infect wild mammals with HPAI via infected bird carcasses (4). Similarly, experimentally infected cats were also susceptible to HPAI H5N1 (A/Vietnam/1194/2004) via infected bird carcasses. The gut, myenteric plexus (5), and olfactory bulb (6) were suggested as potential sites of virus entry into the central nervous system after the consumption of infected birds.

The species barrier between birds and mammals is considerable. Therefore, adaptation of the HPAI virus is needed for efficient replication and transmission in mammals. Current studies suggest at least three requirements for the (efficient) airborne transmission of HPAI viruses between mammals: (i) the efficient attachment of the viral hemagglutinin (HA) glycoprotein to $\alpha$2,6-linked sialic acid receptors. Mutations in HA are required to change the HA binding properties from the avian-type $\alpha$2,3-linked sialic acid receptors to the mammalian-type $\alpha$2,6-linked sialic acid receptor (7, 8). (ii) The optimal stability of the HA protein in mammalian airways. Mutations in HA are required to optimize the fusion of the viral and endosomal membranes as well as the subsequent release of the viral genome in the cytoplasm (9). (iii) Increased virus replication through mammalian adaptation substitutions in the polymerase complex (10, 11). Although these adaptations favored the airborne transmission of the studied HPAI H5N1 strains A/Vietnam/1204/2004 and A/Indo/05/2005, it remains to be elucidated whether these findings can be extrapolated to other clades of HPAI H5N1 viruses. However, mutation E627K was also detected in the polymerase basic protein 2 (PB2) of HPAI H5 clade 2.3.4.4b viruses isolated from mammals in 2021 (3). The PB2-627K variant has been identified as an adaptation of the virus polymerase machinery that likely stimulates virus replication at the lower temperatures of the upper respiratory tract in mammals (10, 11). For HPAI H5 viruses, enhanced virus replication caused by PB2-E627K has been shown to increase pathogenicity *in vitro* and *in vivo* in mice (12–14). Furthermore, the recently discovered mammalian infections of the fox and seals in the UK in late 2020 contained the PB2-D701N adaptation, which was also suggested to increase virus replication (10, 15). These findings may suggest that the current HPAI H5 clade 2.3.4.4b viruses have increased zoonotic potential and are able to infect mammals, resulting in adaptations in the PB2 gene segment.

The previously described cases of HPAI H5 clade 2.3.4.4b infections in seals and foxes as well as the three infected foxes detected in the Netherlands between December of 2021 and February of 2022 suggest that the incidence of infections in mammals might be increasing. In this study, we analyzed virus localization in tissues from the three foxes with related histopathology and showed that the virus is mainly present in the brain, with limited detection in the respiratory tract. A phylogenetic analysis showed that the three fox viruses were not closely related, but they were related to viruses found in wild birds. A genetic analysis demonstrated the presence of mutation PB2-E627K in two out of three fox viruses. Two virus variants were isolated from one of these foxes, with one containing the avian PB2-627E variant and one containing the mammalian PB2-627K variant. The virus replication and stability

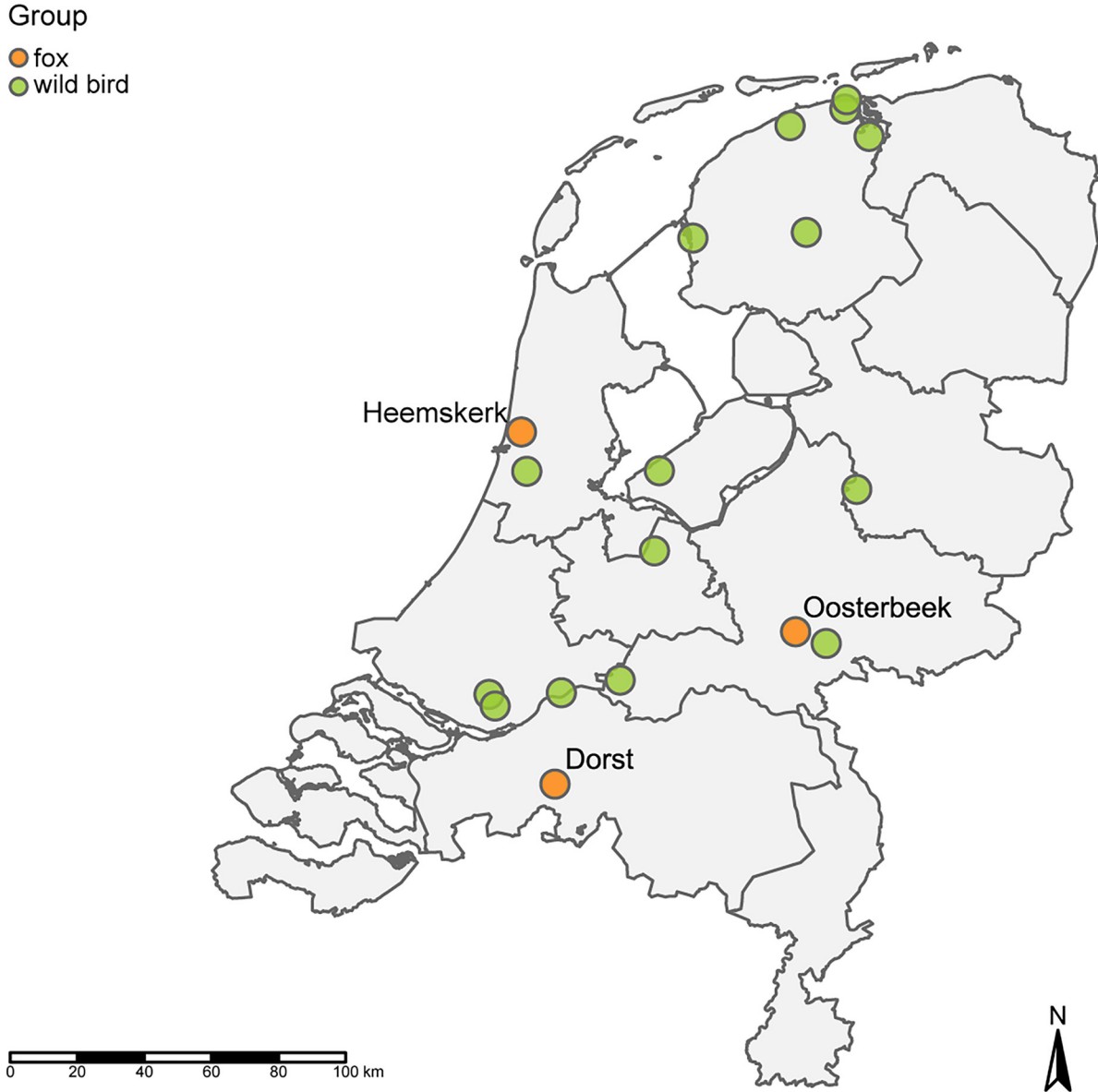

**FIG 1** Locations of the three infected foxes and wild birds from which closely related viruses were isolated during the 2021 to 2022 HPAI H5N1 epizootic in the Netherlands. The map was generated using the R software package tmap (35).

of both isolates was studied over time and showed increased replication for the PB2-627K variant in human and dog cell lines, compared to the chicken cell line, and at the temperature of the mammalian upper respiratory tract, compared to the temperature of the avian respiratory tract, indicating more efficient replication *in vitro* in mammals, compared to birds.

## RESULTS

**Virological analysis of infected foxes.** Three foxes were found in the Netherlands at the municipalities Dorst, Heemskerk, and Oosterbeek, and they were observed to be showing abnormal behavior. The clinical signs in fox-Dorst were apparent blindness, head shaking, falling over, and opisthotonos. This animal was euthanized on 3/12/2021. Fox-Heemskerk showed convulsions, as reported, and was euthanized on 1/1/2022. Fox-Oosterbeek showed lethargy, crouching down with convex back and legs tucked under. The animal lacked fleeing behavior but was alert and did turn its head to look when approached. This fox was humanely dispatched on 7/2/2022. The distance between the different locations was 80 km to 105 km (Fig. 1).

**TABLE 1** Virus detection, hemagglutinin/neuraminidase-subtyping and sequencing[a]

| Sample | Sample type | Ct M PCR | Ct H5 PCR | Ct N1 PCR | PB2-627 consensus[b] | PB2-627 minority[b] | Cleavage site[c] |
|---|---|---|---|---|---|---|---|
| Fox-Dorst | Throat swab | 30.31 | 31.3 | 31.73 | E | K: 37.4% | PLREKRRKR/GLF |
| | Anal swab | 34.95 | 35.57 | 34.78 | ND | ND | ND |
| | Amnion horn and medulla oblongata | 24.13 | 25.81 | ND | E | K: 46.1% | PLREKRRKR/GLF |
| | Cerebrum and cerebellum | 26.23 | 27.19 | ND | E | K: 19.5% | PLREKRRKR/GLF |
| Fox-Heemskerk | Throat swab | 23.33 | 25.06 | 23.84 | E | - | PLKEKRRKR/GLF |
| | Anal swab | no Ct | ND | ND | ND | ND | ND |
| | Amnion horn and medulla oblongata | 23.04 | 25.4 | 24.98 | E | - | PLKEKRRKR/GLF |
| | Cerebrum and cerebellum | 22.19 | 24.09 | 22.78 | E | - | PLKEKRRKR/GLF |
| Fox-Oosterbeek | Throat swab | 26.81 | 26.54 | 27.69 | E | K: 28.8% | PLREKRRKR/GLF |
| | Anal swab | no Ct | ND | ND | ND | ND | ND |
| | Amnion horn and medulla oblongata | 20.99 | 21.88 | 22.93 | ND | ND | ND |
| | Cerebrum and cerebellum | 17.77 | 18.12 | 19.19 | E | - | PLREKRRKR/GLF |

[a]Real-time RT-PCR was used for subtyping; ND, not done.
[b]The PB2-627 consensus and minorities for the avian (E) and mammalian (K) variants were called using Illumina sequencing. -, minority of <1%.
[c]The HA cleavage site sequence was determined via Sanger sequencing.

The foxes were submitted for testing on avian influenza virus. Tracheal and rectal swabs were taken, and brain tissue was collected. Interestingly, the brain samples from two different locations (amnion horn and medulla oblongata, cerebrum and cerebellum) tested positive for avian influenza virus by a quantitative real-time RT-PCR targeting the matrix gene (M-PCR) with high virus genome loads, whereas no virus was detected in rectal swabs, and considerably lower virus loads were detected in the throat swabs of two foxes. Only for fox-Heemskerk were comparable virus loads detected in the brain samples and throat swab (Table 1). The viruses were subtyped as HPAI H5N1 using Sanger sequencing. Virus isolation was performed successfully on the brain tissues of all three foxes, showing the presence of infectious virus in the brain.

**Pathological examination of infected foxes.** All three foxes were adult males with a moderate to poor body condition. Two foxes had a shiny coat, and one fox (Oosterbeek) was covered with mud. The most prominent gross finding was poorly collapsed lungs with a marbled red aspect, which was present with a slight variation in all three foxes. Histology revealed a subacute to chronic purulent granulomatous broncho-interstitial pneumonia with large numbers of parasitic structures (*Angiostrongylus vasorum*) in two of the three foxes. These severe pulmonary changes and the moderate pulmonary changes in the third fox were not associated with virus protein expression (Fig. 2A and B; Table S1). The upper respiratory tract (nasal conchae and trachea) displayed a mild to moderate suppurative inflammation with the presence of parasitic structures (*Capillaria* spp.). In two of the three foxes, the nasal conchae was evaluated (Table S1). In fox-Heemskerk, virus protein expression was mainly observed in the olfactory epithelial cells with associated necrosis of these cells (Fig. 2C), whereas there was no protein expression in the respiratory tract of fox-Oosterbeek. None of the foxes showed virus protein expression in the trachea (Fig. 2D). In all three foxes, strong virus protein expression was present in the brain, which was most prominent in the cerebrum. Virus protein was expressed in neurons and microglia cells in the gray matter, which was associated with a nonsuppurative encephalitis with perivascular cuffing (Fig. 2E and F). There was no virus expression or significant histopathology in the olfactory bulb (Fig. 2G). A subacute lymphoplasmacytic myocarditis with myocardial degeneration and necrosis with virus protein expression in cardiomyocytes was detected only in fox-Heemskerk (Fig. 2H). No virus protein expression was observed in the other investigated organs (intestinal tract [Fig. 2I], pancreas, spleen, liver, and kidney [Table S1]).

**Phylogenetic and genetic analyses of fox viruses.** Full genome sequencing was performed on the brain samples and throat swabs to study the genetic relationship between the viruses. A phylogenetic analysis showed that the viruses belonged to H5 clade 2.3.4.4b and clustered with viruses found in wild birds during the HPAI H5N1 2021 to 2022 epizootic in the Netherlands. The viruses had the same genetic

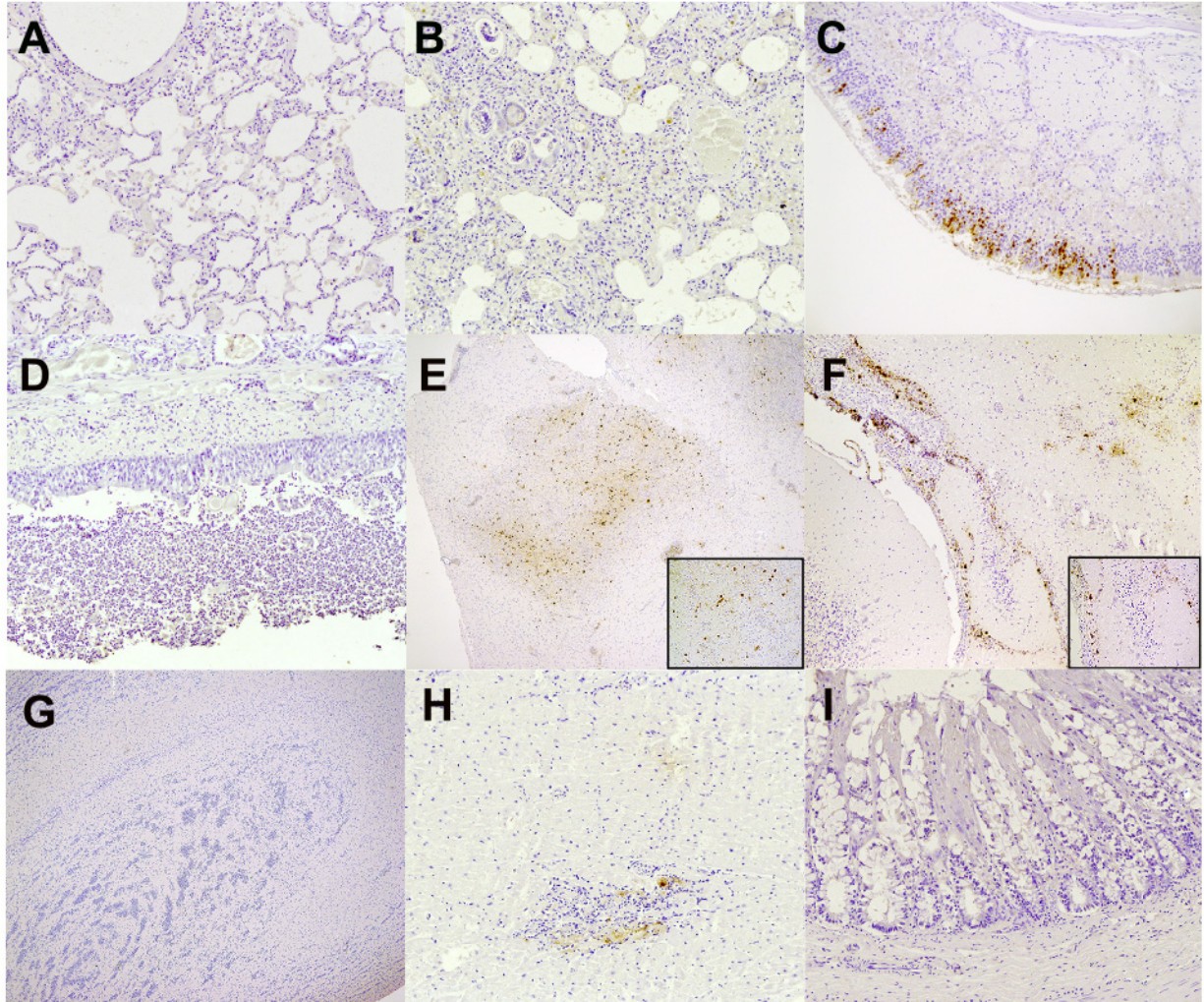

**FIG 2** Histopathology and virus protein expression in tissues of foxes. Immunohistochemistry (IHC) was performed on tissue sections against influenza A NP. The insets show the magnification of IHC stained. (A) Fox-Dorst moderate purulent broncho-interstitial pneumonia (less affected area), no virus protein expression; (B) Fox-Oosterbeek severe purulent broncho-interstitial pneumonia with intralesional larvae (*Angiostrongylus vasorum*), no virus protein expression; (C) Fox-Heemskerk mild to moderate purulent rhinitis with positive staining of olfactory epithelial cells; (D) Fox-Dorst severe necropurulent tracheitis with intralesional parasite eggs (*Capillaria spp*), no virus protein expression; (E) Fox-Dorst cerebrum moderate nonsuppurative encephalitis with virus protein expression in neurons and microglia cells in the neuropil; (F) Fox-Heemskerk cerebellum moderate nonsuppurative encephalitis with virus protein expression in neurons and microglia cells in the neuropil; (G) Fox-Oosterbeek bulbus olfactorius, no significant histopathology or virus protein expression; (H) Fox-Heemskerk moderate lymphoplasmacytic myocarditis with mild myocardial degeneration and necrosis with positive virus protein staining of cardiomyocytes; (I) Fox-Oosterbeek colon, no significant histopathology or virus protein expression. Panels A, B, C, D, H, and I, magnification ×20; panel F, magnification ×10; panels E–G, original magnification 2.5×; insets, magnification ×40.

constitution as did the HPAI H5N1 viruses found in the Netherlands during the 2020 to 2021 epizootic. The closest related wild bird viruses differed between 17 and 42 nucleotide positions from the fox viruses, and they were found between 39 and 149 km distance, 5 to 37 days before the infected foxes were detected (Table S2). The viruses isolated from the foxes were not closely related, based on the phylogenetic analysis, and they differed from each other by between 127 and 176 nucleotides (Fig. 3; Fig. S1). Therefore, the three foxes were likely infected by independent introductions from wild birds.

A genetic analysis was performed to investigate whether the mutations implicated in the adaptations to mammalian hosts occurred in the virus genomes. Mutation screening identified the previously described E627K mutation in the PB2 segment of the fox-Dorst virus. A minority variant analysis of the next-generation sequencing data from three samples

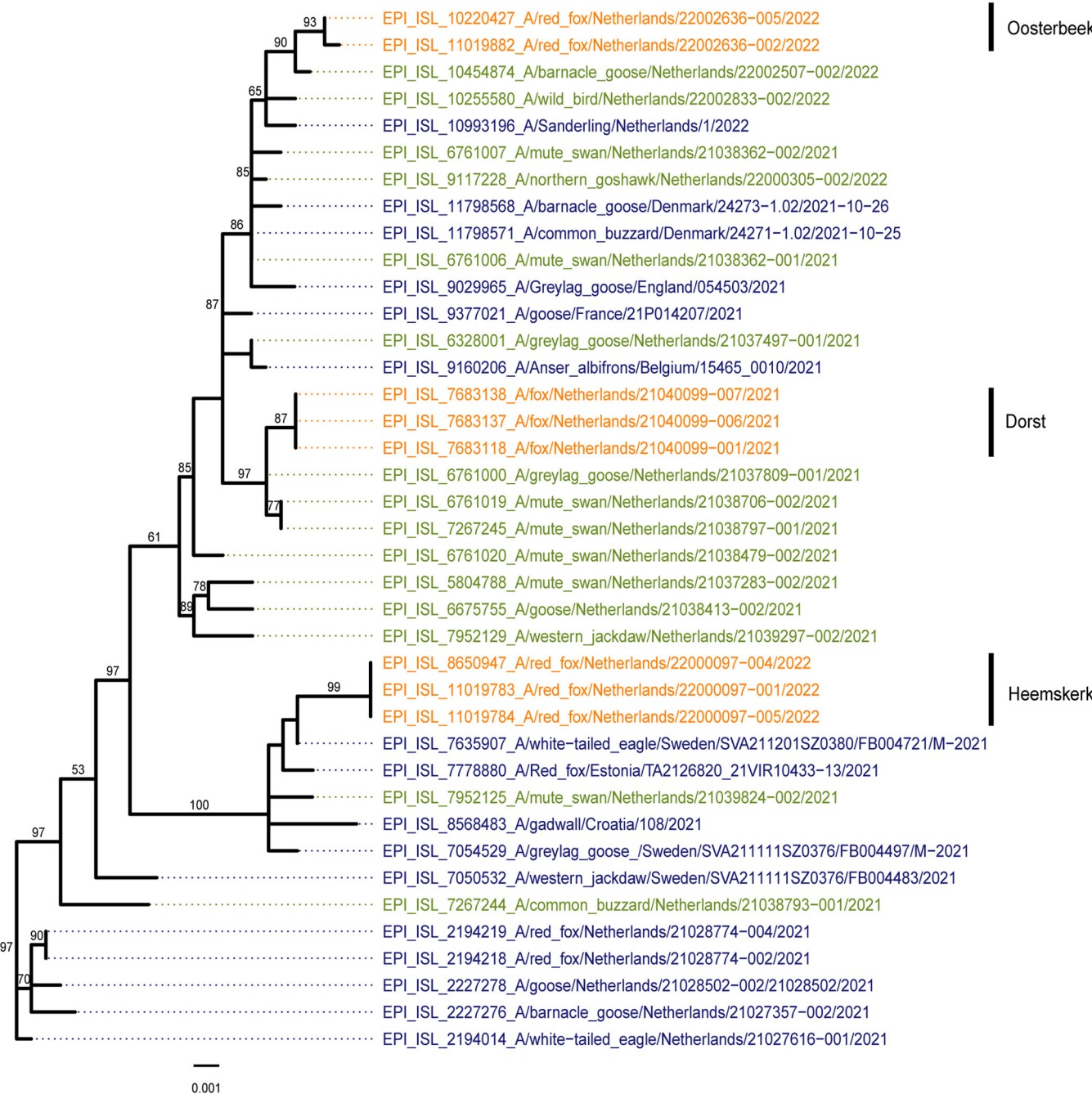

**FIG 3** Phylogenetic tree for the HA segment obtained via the maximum likelihood method that shows the viruses detected in the samples of the three foxes (orange), closely related HA sequences from other viruses detected in the Netherlands (green) and in Europe (blue), and relevant sequences from the 2020 to 2021 epizootic (blue).

of fox-Dorst showed a mixture of the avian (627E) and mammalian (627K) PB2 variants. The percentage of the 627K mutation varied between 37.4% of the virus population in the trachea swab, 46.1% in the amnion horn and medulla oblongata, and 19.5% in the cerebrum and cerebellum (Table 1). Fox-Heemskerk carried only the avian 627E-variant, and fox-Oosterbeek carried the avian 627E-variant with 28.8% of the 627K-variant in the throat swab. Further in-depth analysis of the sequencing data did not reveal other known or previously described host shift adaptations in the fox viruses (results not shown).

**The role of mutation E627K in the replication of the fox-Dorst virus.** A limited dilution series of the amnion horn and medulla oblongata sample derived from fox-

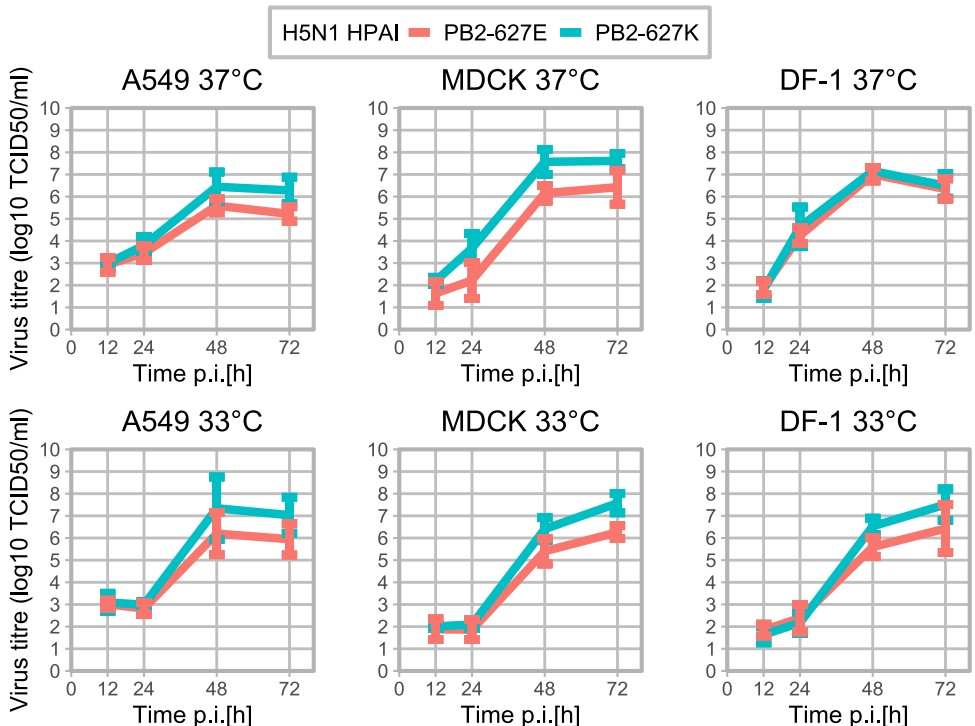

**FIG 4** Virus replication curves of the PB2-627K (blue) and PB2-627E (red) H5N1 HPAI viruses on A549 (human), MDCK (dog), and DF-1 (chicken) cells cultured at 37°C, which is the temperature of the avian upper respiratory tract, and at 33°C, which is the temperature of the mammalian upper respiratory tract. The differences in infectious virus titers between viruses are statistically significant from 48 h p.i. onwards ($P < 0.05$). No significant differences were found between the infectious virus titers of the two viruses on DF-1 cells at 37°C ($P > 0.05$). The virus titers of cells cultured at 33°C instead of 37°C were significantly lower at 24 h p.i. ($P < 0.0001$) and 48 h p.i. ($P < 0.05$), but they were significantly higher at 72 h p.i. ($P < 0.05$).

Dorst was inoculated into embryonated eggs for virus isolation. Full genome sequencing was performed for individual egg isolates. This showed that we isolated a HPAI H5N1 virus with the mammalian PB2-627K variant (100%), and a virus with the avian PB2-627E variant (98,2%) that contained only one additional mutation (G485R) in the nucleoprotein (NP) (99%). To study the effect of PB2-E627K on virus replication, the virus titer was measured at specific time points after the infection of the mammalian A549 and MDCK cell lines as well as the avian DF-1 cell line.

The PB2-E627K mutation appears to increase virus replication from 48h postinfection (p.i.) onwards in mammalian cells (A549, MDCK) but not in avian cells (DF-1) at 37°C (Fig. 4). The putative differences between the two virus variants were statistically assessed using fitted linear mixed models (LMM) with *post hoc* analyses. Due to sample size limitations, three-way interactions containing time, virus, and cell type as variables could not be analyzed. However, the two-way interactions of time with virus and time with temperature could be assessed using the LMM. The differences between the two virus variants were statistically significant from 48 h p.i. onwards ($P < 0.05$) (Table S3). To assess the apparent absence of differences between the PB2-627E and PB2-627K virus replication curves on the DF-1 cells at 37°C, virus titer was compared by performing pairwise comparisons between both viruses at each time point and ignoring the data dependency (no random effects introduced in the model) due to replicated measures. These pairwise comparisons showed no significant differences ($P > 0.05$) between the two viruses from 48 h p.i. onwards. These results may indicate that PB2-E627K enhances virus replication in mammalian cells but not in avian cells at the temperature of the avian upper respiratory tract (37°C).

The PB2-E627K mutation is likely an adaptation of the polymerase to the lower body temperature of mammals, compared to that of birds. Therefore, virus replication was also studied at 33°C, which is the approximate temperature of the mammalian

**TABLE 2** Serial passaging of the PB2-627 variants on cell lines[a]

| Passage | 1 | 2 | 3 | 4 | 5 | 6 | 7 | 8 | 9 | 10 |
|---|---|---|---|---|---|---|---|---|---|---|
| A549 37°C | 9.4% | 80.7% | 81.7% | 94.5% | 98.6% | 100% | 100% | 100% | 100% | 100% |
| MDCK 37°C | 18.1% | 49.7% | 95.2% | 97.8% | 99.0% | 100% | 100% | 100% | 100% | 100% |
| DF-1 37°C | 32.6% | 87.3% | 93.1% | 91.4% | 93.8% | 94.4% | 97.2% | 97.9% | 98.8% | 98.3% |
| A549 33°C | 13.8% | 44.9% | 94.0% | - | 100% | - | - | - | 100% | 100% |
| MDCK 33°C | 26.8% | 91.7% | 98.4% | 100% | 100% | 100% | 100% | 100% | 100% | 100% |
| DF-1 33°C | 77.9% | 97.7% | 100% | 100% | 100% | 100% | 100% | 100% | 100% | 100% |

[a]Percentage of reads containing the mammalian PB2-627K variant over serial passages on A549, MDCK, and DF-1 cells at 37°C and 33°C. The minority threshold is 1%. -, the sample did not pass the quality criteria.

upper respiratory tract. The LMM analysis of the variables time and temperature indicated that virus replication was significantly lower at 24 h p.i. ($P < 0.0001$) and 48 h p.i. ($P < 0.05$) but significantly higher at 72 h p.i. ($P < 0.05$), indicating that the replication of both virus variants is delayed but not impaired (Fig. 4; Table S4). In contrast with our findings at 37°C, the virus titers were significantly higher for the PB2-627K variant than for the PB2-627E variant on DF-1 cells cultured at 33°C (Fig. 4; Table S3). Thus, PB2-E627K increases virus replication in mammalian cells and avian cells at the temperature of the upper respiratory tract of mammals (33°C).

**Serial passaging of the two virus variants.** Serial passages were performed on the mammalian A549 and MDCK cell lines as well as the avian DF-1 cell line cultured at 37°C and 33°C to assess the stability of the obtained virus isolates. The 100% isolate of PB2-627K was stable on all experimental conditions for 10 passages in all cell lines at both temperatures tested (data not shown). However, the passaging of the PB2-627E virus isolate, which contains 1.8% PB2-627K, showed that the 627K mutant has a replication benefit in all three cell lines at both temperatures tested (Table 2). However, whereas the 627K-mutation is selected in the mammalian cell lines within six passages at 37°C, selection in the avian DF-1 cell line takes more than 10 passages. These results confirm our previous findings that the PB2-E627K mutation increases the replication capacity of the virus in mammalian cells but not in avian cells. Furthermore, PB2-627K was selected faster (within 3 or 4 passages) on cells incubated at 33°C, compared to cells incubated at 37°C. Thus, the lower temperatures of the mammalian upper respiratory tract (33°C) could be involved in the selection of the PB2-E627K mutation.

## DISCUSSION

Here, we investigated three cases of HPAI H5N1 infections in wild foxes in the Netherlands to assess tissue tropism in wild mammals and to screen for adaptive mutations. All three foxes showed unusual tissue tropism, and evidence was found for mammalian adaptation. The full genome sequencing of the fox viruses, followed by a phylogenetic analysis, demonstrates that they belong to clade 2.3.4.4b and are related to viruses detected in wild birds. The fact that the three fox-viruses were not closely related as well as the large distance between the locations at which the three foxes were found suggest that these were separate virus introductions that likely originated from wild birds. Experimental infections via the consumption of infected bird carcasses have indicated that foxes are susceptible to avian influenza virus (AIV) infections, and this is also the most probable route of infection for free-living foxes (4). Virus RNA was most abundant in the brains of all three foxes, and it was associated with positive IHC staining for virus protein in the brain tissue of all three foxes. This finding is novel, as previous HPAI H5 virus infections in foxes showed a diffuse tropism with high virus replication in the respiratory system, similar to HPAI infections in poultry (1, 4). It is unclear how the virus infected the brain without clear viremia and systemic replication. Fox-Heemskerk showed mild positive staining by IHC of the olfactory epithelium of the nasal conchae. We also detected high viral RNA loads in the throat swab of this fox, and these most likely originated from the olfactory epithelium, as all other respiratory organs were tested negative by IHC. The olfactory epithelium is connected to the brain via the

olfactory bulb and has previously been described as a point of entry for AIV (5, 6). However, we found no evidence of AIV replication in the olfactory bulb, based on IHC and the absence of histopathologic changes, whereas in other parts of the brain, virus expression was always associated with histopathologic changes. The olfactory epithelium could play a role in virus entry, but the exact route of infection remains to be elucidated. Fox-Heemskerk also showed mild positive virus staining of cardiomyocytes. The infection of cardiomyocytes normally indicates a systemic disease and viremia. Currently, we have no information on the early stages of infection or on infections that may occur without the display of neurological symptoms in wild foxes. Thus, we cannot exclude the possibility that virus replication also occurred in the respiratory tract or other organs at earlier stages of the infection. However, in these foxes, which showed neurological symptoms, the virus was found in the brain, suggesting a strong neurotropism of the virus. Neurological symptoms have also been reported for avian species. For example, HPAI disease in chickens is short and results in sudden death; however, in domestic ducks and wild birds, the disease is longer, and neurological symptoms, such as partial paralysis and tremors, can often be observed (16–18). However, HPAI is generally considered to be a respiratory disease with a high virus genomic RNA concentration in the respiratory system (19). Thus, the lack of virus replication in the respiratory system of the wild foxes is interesting, and the detection in the brain suggests that the current HPAI H5N1 2021 to 2022 viruses have a strong neurotropism in mammals.

In two of the three foxes, a minority population of viruses that contained the zoonotic mutation PB2-E627K was identified. The fact that this mutation was not detected in any of the wild bird sequences during the 2021 to 2022 epizootic in the Netherlands suggests that this mutation quickly arises upon the infection of mammals. Furthermore, as both the PB2-627E and PB2-627K variants were detected in these two foxes, it appears likely that this mammalian adaptation emerged within these specific animals. A similar event has occurred in two seals infected with the HPAI H5N8 virus in August of 2021 in Germany, and it indicates selection pressure for this adaptation in mammals (3). Histopathology revealed distinct virus protein expression and associated brain histopathology with no virus replication found by IHC in the lungs of these seals, similar to those of the foxes that were investigated in this study. We also showed that mutation PB2-E627K increased replication in the mammalian cell lines at both 33°C and 37°C, whereas for the avian DF-1 cell line, this was not observed at 37°C. Furthermore, in passaging experiments, the mutation PB2-627K was found to have a strong replication benefit in the tested mammalian cell lines at 37°C, whereas this effect was smaller on the avian cell line at this temperature. At 33°C, the mutation PB2-627K was found to increase the replication capacity of the virus in all three cell lines to similar levels. Thus, the mutation PB2-E627K increases the replication speed in mammalian cells, but not in avian cells, at the relevant temperature of the upper respiratory tract (33°C). This finding is supported by previous reports on the PB2-E627K mutation (10, 11). Therefore, the lower temperatures of the mammalian upper respiratory tract could be a driving factor of the emergence of the PB2-E627K mutation. Although PB2-E627K improves virus replication in mammalian cells, the mutation appears inessential for virus replication in mammalian cells, which is in agreement with the initial introduction of the HPAI H5N1 virus carrying the PB2-627E variant from wild birds. The additional mutation G485R in the NP that is present in the isolated PB2-627E virus could be associated with the adaptation to mammals, as described in a previous study (20); however, its effect on virus replication is not known. Although a relatively small increase in replication capacity was measured in mammalian cells due to the PB2-E627K mutation, previous studies have indicated that small increases in HPAI virus replication coincided with increased pathogenicity in mammals (12, 13). Increased virus replication may also stimulate the emergence of further mammalian adaptations, as more genomic copies are produced per introduction.

It is currently unclear which factors have contributed to the increase of infections observed in red foxes in nature. The HPAI H5N1 clade 2.3.4.4b virus may be more capable of infecting mammals, the virus may be more infectious, or there may be a higher

prevalence of the virus in wild birds during the 2021 to 2022 epizootic, compared to previous epizootics. Limited virus shedding and virus replication were observed in the respiratory systems and digestive tracts of the investigated foxes, which likely limits transmission between mammals at this state of the disease. Consistent with this, no evidence for transmission between foxes was found, based on the phylogenetic analysis of the viruses. The genetic analysis suggests that the zoonotic mutation PB2-E627K may arise in infected mammals. Although PB2-E627K is an important mammalian adaptation, previous research has indicated that several further adaptations are required for the efficient airborne transmission of AIV between mammals. For example, the PB2-E627K, HA-Q222L, and HA-G224S mammalian adaptations were required before 10 serial passages in ferrets to eventually produce airborne variants of HPAI (21). In particular, adaptations in the viral HA glycoprotein, which affects the stability in the mammalian airways and the receptor binding specificity to the $\alpha$2,6-linked sialic acid receptors in the mammalian upper respiratory tract, should be monitored closely to prevent potential spread between mammals. Unfortunately, infections of mammalian carnivorous wildlife are difficult to prevent during HPAI epizootics when large numbers of wild birds are affected. The clearing of wild bird carcasses could help limit these HPAI introductions into wild mammals, but this may not be feasible during large outbreaks or at remote sites in nature. Awareness should be raised for the potential transmission of HPAI viruses from wild birds to pet animals. Dogs and cats may be at risk when interacting with or feeding on infected wild birds or their carcasses. The observed tissue tropism in these foxes and the lack of evidence for further spreading between wild mammals indicates that it is unlikely that the HPAI H5N1 clade 2.3.4.4b spreads to humans. However, surveillance for HPAI viruses in wild mammals should be expanded to closely monitor potential transmission between mammals and the emergence of zoonotic mutations for pandemic preparedness.

## MATERIALS AND METHODS

**Tissue sampling, virus detection, and histopathology.** Three foxes showing neurologic signs were submitted for necropsy to exclude rabies and influenza A virus infection, according to governmental surveillance guidelines. A postmortem examination was performed within 2 days after euthanasia. Various tissue samples were taken for histopathology and immunohistochemistry, and they were fixed in 10% neutral buffered formalin. The tissues were processed and evaluated for histopathologic changes with hematoxylin and eosin stain (H&E) and for influenza A NP expression with IHC, as described previously (16). From all animals, an anal swab, a throat swab, and brain sections (amnion horn and medulla oblongata, cerebrum and cerebellum) were collected. Swabs collected during the postmortem examination were placed in 2 mL of tryptose phosphate broth supplemented with 2.95% gentamicin. Avian influenza virus RNA was subsequently extracted using a MagNA Pure 96 system (Roche, Basel, Switzerland). AIV was detected by a quantitative real-time RT-PCR targeting the matrix gene (M-PCR), as described previously (22). Positive samples were subtyped using H5-specific and N1-specific real-time RT-PCRs, as recommended by the European Union Reference Laboratory (23, 24). For at least one sample of each fox, the HA cleavage site sequence and the N subtype were determined via Sanger sequencing, as described previously (22).

**Complete genome sequencing and analysis.** All virus genome sequences were determined directly on the swab or tissue samples. The virus RNA was purified using a High Pure Viral RNA Kit (Roche, Basel, Switzerland), amplified using universal eight-segment primers, and directly sequenced, as described previously (22). Purified amplicons were sequenced at high coverage (average of >1,000 per nucleotide position), using the Illumina DNA Prep method and Illumina MiSeq 150PE sequencing. The reads were mapped using the ViralProfiler-Workflow, an extension of the CLC Genomics Workbench (Qiagen, Germany). Consensus sequences were generated by a reference-based method. Reads were first mapped to a reference set of genomes, and they were subsequently remapped to the closest reference sequence. Finally, the consensus sequence of the complete virus genome was extracted, and the minority variants were called using a cutoff of 1%. The fox viruses were analyzed using FluServer for mutations that increased the zoonotic potential of the virus (25).

**Phylogenetic analysis.** In addition to the virus sequences obtained from foxes and wild birds in the Netherlands in this study, the top five BLAST results for related sequences in Eurasia were included in the phylogenetic analysis. These H5N1 genome sequences were downloaded from the GISAID database (26). A phylogenetic analysis of the complete genome sequences was performed for each genome segment separately, aligning the virus sequences using MAFFT v7.475 (27) and reconstructing the phylogeny using a maximum likelihood (ML) analysis with IQ-TREE software v2.0.3 and 1,000 bootstrap replicates (28). The ML tree was visualized using the R package ggtree (29). The GISAID sequences used in the phylogenetic analysis are listed in Table S5, in which we acknowledge all of the contributors to the GISAID database.

**Cell cultures.** Madin-Darby canine kidney (MDCK) cells obtained from Philips-Duphar (Weesp, the Netherlands), chicken embryo fibroblasts (DF-1) (ATCC, Wesel, Germany), and human lung alveolar epithelial cells (A549) (ATCC, Wesel, Germany) were maintained in cell culture medium consisting of Dulbecco's Modified Eagle Medium GlutaMAX (DMEM) (Thermo Fisher Scientific, Bleiswijk, Netherlands) supplemented with 5% fetal calf serum (Capricorn scientific, Germany) and 0.1% penicillin-streptomycin (Thermo Fisher Scientific, Bleiswijk, Netherlands) at 37°C and 5% $CO_2$. The cells were passaged when confluent using 0.05% Trypsin-EDTA (Thermo Fischer Scientific, Bleiswijk, Netherlands).

**Virus isolation, titration, and propagation.** Homogenates from the brain tissue (amnion horn and medulla oblongata, cerebrum and cerebellum) were incubated for 1 h with 1% penicillin and 1% gentamicin at room temperature. The homogenates were filtered and injected in nine-day-old specific-pathogen-free (SPF) embryonated chicken eggs (ECE), as described previously (30). Allantoic fluid was harvested from the deceased eggs, aliquoted, stored at −80°C, and sequenced using Illumina sequencing, as described above. Subsequently, a serial dilution of the amnion horn and medulla oblongata homogenate was injected in fresh nine-day-old SPF ECEs. Allantoic fluid was harvested from all eggs and individually sequenced. The median tissue culture infective dose (TCID50) of the isolated viruses was determined via endpoint titration on MDCK cells. In short, $2.5 \times 10^4$ cells per well of a 96-well plate were seeded overnight in cell culture medium. Monolayers were infected with 10-fold serial dilutions in infection medium consisting of DMEM GlutaMAX supplemented with 0.1% penicillin-streptomycin and 0.3% bovine serum albumin. Each dilution was tested in eightfold, and each titration was diluted in triplicate. After 2 days of incubation at 37°C and 5% $CO_2$, the monolayers were stained via immunoperoxidase monolayer assay. The monolayers were fixed with 10% neutral buffered formalin. The primary antibody was produced in-house from HB 65 mouse anti-NP and diluted 1:2,500. This was followed by HRP-conjugated rabbit anti-mouse that was diluted 1:500 (Dako, Glostrup, Denmark). The titration was repeated on a different day, and the TCID50 titers were calculated using the Reed and Muench method (31).

**Virus replication.** The multiplicity of infection (MOI) was optimized for each cell line by selecting a dilution, which caused limited cell death but sufficient virus replication. MDCK, DF-1, and A549 cells were seeded in duplo on a 24-well plate at a density of $2.5 \times 10^5$ cells per well in cell culture medium. The next day, the cells were infected with 1.5 mL of virus diluted in infection medium at a multiplicity of infection of 0.01 (A549), 0.001 (MDCK), or 0.0005 (DF-1). The cells were incubated at 37°C and 5% $CO_2$ or at 33°C and 5% $CO_2$. At the indicated time points, 150 $\mu$L of medium was harvested and stored at −80°C. 150 $\mu$L of fresh infection medium was added after each harvest. The infectious virus titer of the collected medium was determined in triplo using the TCID50 protocol described above.

**Passaging of viruses and sequence analysis.** The stability of the virus isolates was determined by passaging 10 times on A549, MDCK, and DF-1 cells at identical temperatures and multiplicities of infection as the replication experiment described above. In short, the cells were seeded at a density of $2.5 \times 10^6$ cells per T-25 flask. The next day, the cells were infected with diluted virus in 3 mL infection medium. Cells were incubated either at 37°C and 5% $CO_2$ or at 33°C and 5% $CO_2$ for 3 days before the medium was collected. The medium was diluted 1,000× and added to a fresh T-25 flask of A549, MDCK, or DF-1 cells. The remaining, undiluted medium was stored at −80°C for sequencing. The viral RNA was isolated using a Zymo Quick-RNA Viral 96 Kit (BaseClear, Leiden, Netherlands). A region of 168 bp on the PB2 protein was amplified via RT-PCR using custom primers (Table S6) (Eurogentec, Maastricht, Netherlands) and a Qiagen OneStep RT-PCR Kit (Qiagen, Venlo, Netherlands). The amplicon length and concentration were analyzed using HS DNA 1000 with TapeStation 2200 (Agilent) as well as Quanti-IT (ThermoFisher) with ClarioStar (BMG). Region-specific amplicons were individually barcoded using Illumina's Nextera XT Index V2 Kit with limit cycle PCR and PE150 sequenced on Illumina's Miseq. The reads were mapped using the ViralProfiler-Workflow, an extension of the CLC Genomics Workbench (Qiagen, Germany). Consensus sequences were generated via a reference-based method, and minority variants were called using a cutoff of 1%. The minimal coverage at position PB2-627K was 2,000 reads.

**Statistics.** For the statistical analyses of the virus replication curves, we considered two levels of dependency in the generated data: repeated measures nested within a replicate and replicate nested within a virus-cell combination. To account for these dependencies, we fitted linear mixed models (LMM) in which this nested structure and the repeated measures were included as random effects. The virus titer was the response variable. The variables time, virus (two isolates: PB2-627E and PB2-627K), cell (MDCK, A549, or DF-1), and temperature (37°C or 33°C), as well as their interactions, were assessed to evaluate their significance. The significant interactions ($P < 0.05$) kept in the final model were time:virus, time:temperature, and time:cell. Variable significance was assessed using the likelihood ratio test. This analysis was done using the statistical software R (32). The LMM was fitted using the R package lmer (33), and the *post hoc* tests for the pairwise comparisons were done using the R package emmeans (34). Further assessments were done for the data generated from the cell line DF-1. Because of the limited number of observations, this analysis was done by fitting linear models and assuming that the observations were independent (see Results).

**Data availability.** The virus sequences generated in this study were submitted to the GISAID database, and the accession numbers are listed in Table S2. The virus isolates from the cerebrum and cerebellum include A/red fox/Netherlands/21040099-007/2021 (fox-Dorst), A/red fox/Netherlands/22000097-005/2022 (fox-Heemskerk), and A/red fox/Netherlands/22002636-005/2022 (fox-Oosterbeek).

## SUPPLEMENTAL MATERIAL

Supplemental material is available online only.

**SUPPLEMENTAL FILE 1**, PDF file, 0.3 MB.

## ACKNOWLEDGMENTS

We acknowledge Albert G. de Boer, Arno-Jan Feddema, Corry H. Dolstra, Eline Verheij, and Frank Harders for their technical assistance. We acknowledge Latoya Siemons VRC Zundert, St. Dierenambulance Kennemerland, Dierenkliniek Castricum, Faunabeheer Middachten, and the Netherlands Food and Consumer Product Safety Authority (NVWA) for notifying and submitting the foxes. We thank Wim H. M. van der Poel for the critical reading of the manuscript. This research was funded by the Dutch Ministry of Agriculture, Nature, and Food Quality (project WOT-01-003-096 and KB-37-003-039).

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
