## [Reviewer comments · Microbiology Spectrum]

Microbiology Spectrum

Highly pathogenic avian influenza H5N1 virus infections in wild red foxes (*Vulpes vulpes*) show neurotropism and adaptive virus mutations

Luca Bordes, Sandra Vreman, Rene Heutink, Marit Roose, Sandra Venema, Sylvia Pritz-Verschuren, Jolianne Rijks, Jose Gonzales, evelien germeraad, Marc Engelsma, and Nancy Beerens

Corresponding Author(s): Nancy Beerens, Wageningen Bioveterinary Research

Review Timeline:

Submission Date:	July 25, 2022
Editorial Decision:	October 3, 2022
Revision Received:	November 9, 2022
Editorial Decision:	December 1, 2022
Revision Received:	December 21, 2022
Accepted:	December 23, 2022

Editor: Mathilde Richard

Reviewer(s): The reviewers have opted to remain anonymous.

Transaction Report:

DOI: <https://doi.org/10.1128/spectrum.02867-22>

October 3, 2022

Dr. Nancy Beerens
Wageningen Bioveterinary Research
Houtribweg 39
Lelystad
Netherlands

Re: Spectrum02867-22 (Highly pathogenic avian influenza H5N1 virus infections in wild red foxes (*Vulpes vulpes*) show neurotropism and adaptive virus mutations)

Dear Dr. Nancy Beerens:

Thank you for submitting your manuscript to Microbiology Spectrum. It has been reviewed by two experts in the field. While they acknowledged the potential interest of your study for the readers of Microbiology Spectrum, they also have raised points of concerns (see below for the full reviewers' reports). Please address all the comments raised by the reviewers, especially those from Reviewer 1 regarding conclusions and extrapolations of the data to the transmissibility and neurotropism phenotypes of the described viruses.

Link Not Available

Sincerely,

Mathilde Richard

Journals Department
Reviewer comments:

Reviewer #1 (Comments for the Author):

The manuscript by Bordes, et al. describes postmortem pathology and virology of HPAI H5N1 viruses from three wild red foxes in the Netherlands from December 2021-February 2022. This is an interesting and well-done study. Some comments for further

consideration by the authors are listed below:

1. Was an attempt to culture virus from all 3 foxes or only from Fox-Dorst as described in the manuscript?
2. It is interesting that Fox-Heemskerk H5N1 sequence analysis only revealed PB2-E627 with no evidence of E627K mutation and yet also was associated with CNS and myocardial viral infection post-mortem. It would be interesting to know if viral isolates from Fox-Heemskerk were available for the comparative *in vitro* growth studies described with viruses from Fox-Dorst.
3. While it is of course understandable that no data are available from these wild foxes on the duration of their infection and clinical illness, it is still interesting that moderate levels of virus (at least by RT-PCR) were present in throat swabs. It would be really important in future such studies if attempts at viral titers were performed from upper respiratory samples. Given the lack of detection of viral protein in respiratory tissues by immunohistochemistry, how do the authors explain the levels of viral RNA in the throat swabs of fox-Heemskerk and fox-Oosterbeek, especially? Do you think infectious virus is present in the throat or only viral RNA? If yes, do you think this level of virus is potentially transmissible? Is it possible to perform RT-PCR for viral RNA from the fixed respiratory tissues? Perhaps viral replication at time of euthanasia in lung was lower than detectable by IHC.
4. It would be very interesting to see comparative infection and pathogenesis studies done in mice with at least the Fox-Dorst PB2 E627 and K627 H5N1 viruses.
5. It would be helpful for the authors to list all the SNPs (especially the nonsynonymous SNPs in a supplementary table against a reference H5N1. While not a part of this study directly, were any gallinaceous poultry samples in the 2021-2022 H5N1 epizootic uploaded to databases sequence positive for the PB2 E627K mutation?

Reviewer #2 (Comments for the Author):

The paper by Border et al describes the detection of HPAI H5N1 viruses in foxes in The Netherlands and detected the E627K mutation in 2 out of 3 foxes. This manuscript contains important information on the ability to spread and cause disease in mammals as well as its potential to acquire amino acid substitutions that are associated with increased replication in mammals. However, the data do not support all of their conclusions. First of all, it is unclear when these foxes were exposed to HPAI H5N1 virus and thus what happened in the early phases of the infection. The absence of virus in the respiratory tract, might be due to the fact that replication in the respiratory tract has already peaked (like observed in experimentally inoculated foxes and ferrets). Therefore, it is not possible to say anything about the transmissibility of this virus in foxes. Second, the fact that the virus is mainly detected in the CNS of these foxes at the moment they were euthanized, does not make it more neurotropic than other H5 viruses.

Specific comments:

Line 85-86: This is indeed true for humans, but not all mammals express predominantly 2,6-linked sialic acids in the upper respiratory tract. In e.g. horses and seals 2,3-linked sialic acids are more common in the upper respiratory tract.

Lines 92-93: This has not only been studied for A/Vietnam/1204/2004, but f.e. also for A/Indo/05/2005 (PMID: 24725402).

Line 122-129: Where all three foxes euthanized on the day they were found in the wild? If these foxes were first placed in a rehabilitation center please add how long the foxes have been kept, and the progression of the disease in time.

Line 141-160: Please describe consistently the histopathological changes at the different anatomical locations of the respiratory tract of the individual animals and whether these were associated with virus antigen. For example, virus antigen was observed in the olfactory mucosa of 1 fox, but it is unclear if histological lesions or virus antigen were observed in the olfactory mucosa of other foxes, or whether the olfactory mucosa was not studied for the other foxes. In addition, there seem to be some discrepancies in table S1. For Fox-Dorst nasal concha were negative for virus antigen, but histopathologic changes were not analyzed. Same for trachea of Fox-Heemskerk. I assume if the tissues are collected both analyses can be performed.

Line 155: Why do the authors refer to the histological changes of poliоencephalitis? What are the differences compared to other viral encephalitis.

Line 174-175: it is unclear for which mutations was screened, please include.

Line 196: virus titers already peak at 48 hours post infection in mammalian cells.

Line 198-210: It is unclear what is tested for Table S3, it seems that for only one cell line this has been tested for the different time points. And why could this method not be used for the DF-1 cells?

Line 223-235: It is unclear if the virus passaging was performed multiple times? If this has only been performed once, the only conclusion that can be drawn from it is that E672K has an increased fitness in all cell lines.

Line 247-250. As there is no information on when these foxes got infected it might be that virus replication in the respiratory tract has already peaked, and is therefore lower in the brain. As previous studies in foxes have shown, virus replication peaks early after exposure (PMID: 19046504). In ferrets similar kinetics are observed.

Line 264-265: As the authors already mention before, it is unclear what happened in earlier stages of the infections, so you cannot conclude this virus is exclusive neurotropic. It is very likely that virus replication in the respiratory tract peaked early in the infection and is therefore not detected in the respiratory tract any more.

Line 270-272: This is an overstatement.

Line 283-296: The in vivo experiments show that both virus replicate very efficient in all cell lines at both temperatures. It might be that in some cells the viruses with E726K replicate somewhat faster, but this observation is not solid enough to extrapolate these finding to the in vivo situation in foxes (a quick google search showed that the body temp of foxes is 38.7 degrees Celsius).

Line 308: How does an increased neurotropism contributes to the infection of wild red foxes?

Line 308-310: The fact that at the end-stage of disease virus is not detected in the respiratory tract, or the absence of evidence of virus spread among these 3 individuals, does not mean that this virus is not transmitted among foxes. For such conclusions other studies have to be performed (active surveillance, serology etc).

Staff Comments:

Preparing Revision Guidelines

Please return the manuscript within 60 days; if you cannot complete the modification within this time period, please contact me. If you do not wish to modify the manuscript and prefer to submit it to another journal, please notify me of your decision immediately so that the manuscript may be formally withdrawn from consideration by Microbiology Spectrum.

Reviewer #1 (Comments for the Author):

The manuscript by Bordes, et al. describes postmortem pathology and virology of HPAI H5N1 viruses from three wild reed foxes in the Netherlands from December 2021-February 2022. This is an interesting and well-done study. Some comments for further consideration by the authors are listed below:

Answer: We thank this reviewer for careful reading of the manuscript, and for the compliments on the study. The individual points raised by this reviewer are answered below.

1. Was an attempt to culture virus from all 3 foxes or only from Fox-Dorst as described in the manuscript?

Answer: Viruses were isolated for all three foxes upon inoculation of brain samples into embryonated chicken eggs, showing the presence of infectious virus (lines 138-140). Only for Fox-Dorst a limited dilution series was initiated in eggs to separate the two virus variants detected in the brain of this fox (lines 191-192).

2. It is interesting that Fox-Heemskerk H5N1 sequence analysis only revealed PB2-E627 with no evidence of E627K mutation and yet also was associated with CNS and myocardial viral infection post-mortem. It would be interesting to know if viral isolates from Fox-Heemskerk were available for the comparative in vitro growth studies described with viruses from Fox-Dorst.

Answer: The mutation PB2-E627K is most likely not responsible for the distinct CNS and mild myocardial viral infection. In fox-Dorst a mixture of E627 and K627 was observed in the brain, suggesting that this mutation emerged in this animal after infection. This suggests that fox-Dorst was initially infected by a wild bird variant of the virus carrying the avian E627 amino acid, as was fox-Heemskerk.

3. While it is of course understandable that no data are available from these wild foxes on the duration of their infection and clinical illness, it is still interesting that moderate levels of virus (at least by RT-PCR) were present in throat swabs. It would be really important in future such studies if attempts at viral titers were performed from upper respiratory samples. Given the lack of detection of viral protein in respiratory tissues by immunohistochemistry, how do the authors explain the levels of viral RNA in the throat swabs of fox-Heemskerk and fox-Oosterbeek, especially? Do you think infectious virus is present in the throat or only viral RNA? If yes, do you think this level of virus is potentially transmissible? Is it possible to perform RT-PCR for viral RNA from the fixed respiratory tissues? Perhaps viral replication at time of euthanasia in lung was lower than detectable by IHC.

Answer: The detection limit of PCR is higher than that of IHC. Therefore, some samples that test positive by PCR (with high Ct values) may test negative by IHC. PCR will detect viral RNA and does not provide information on the amount of infectious virus. As the lung samples were fixed, we cannot test these for the presence of infectious virus anymore, and we have no validated assays for PCR on fixated tissues. In future studies, we may consider testing lung tissue directly (before fixation) for the presence of infectious virus.

4. It would be very interesting to see comparative infection and pathogenesis studies done in mice with at least the Fox-Dorst PB2 E627 and K627 H5N1 viruses.

Answer: We fully agree these variants would be interesting to study in infection studies in mammals, however this is beyond the scope of this field study.

5. It would be helpful for the authors to list all the SNPs (especially the nonsynonymous SNPs in a supplementary table against a reference H5N1. While not a part of this study directly, were any gallinaceous poultry samples in the 2021-2022 H5N1 epizootic uploaded to databases sequence positive for the PB2 E627K mutation?

Answer: We added a list of amino acids which differ between the fox viruses and the closest wild bird virus sequence found in the Netherlands to Supplemental table 2. No avian species (including

gallinaceous poultry) were found to be infected with the PB2-627K variant in the Netherlands during the 2020-2021 outbreak.

Reviewer #2 (Comments for the Author):

The paper by Border et al describes the detection of HPAI H5N1 viruses in foxes in The Netherlands and detected the E627K mutation in 2 out of 3 foxes. This manuscript contains important information on the ability to spread and cause disease in mammals as well as its potential to acquire amino acid substitutions that are associated with increased replication in mammals. However, the data do not support all of their conclusions. First of all, it is unclear when these foxes were exposed to HPAI H5N1 virus and thus what happened in the early phases of the infection. The absence of virus in the respiratory tract, might be due to the fact that replication in the respiratory tract has already peaked (like observed in experimentally inoculated foxes and ferrets). Therefore, it is not possible to say anything about the transmissibility of this virus in foxes. Second, the fact that the virus is mainly detected in the CNS of these foxes at the moment they were euthanized, does not make it more neurotropic than other H5 viruses.

Answer: We thank the reviewer for careful reading of the manuscript, and for the constructive suggestions to improve the manuscript. We do realize this field study is limited to the foxes that were found with neurological symptoms in the wild, which represent end stages of the infection. Infection studies will have to be performed to study earlier stages of the infection, but these are beyond the scope of this field study. However, to our knowledge, no infection studies with H5-viruses showed neurotropism as strongly combined with limited expression of viral antigen in the respiratory tract as observed in our field study (red foxes PMID: 19046504, mice PMID: 28873012, cats PMID: 16400021 and PMID: 18619884).

Specific comments:

Line 85-86: This is indeed true for humans, but not all mammals express predominantly 2,6-linked sialic acids in the upper respiratory tract. In e.g. horses and seals 2,3-linked sialic acids are more common in the upper respiratory tract.

Answer: Indeed 2,6 and 2,3-linked sialic acids are both present in mammals and avian species in a variety of distributions also depending on the location in the respiratory tract. However, a switch from 2,3 to 2,6-linked sialic acids is commonly accepted by the field as an adaptation to mammals, and humans in particular.

Lines 92-93: This has not only been studied for A/Vietnam/1204/2004, but f.e. also for A/Indo/05/2005 (PMID: 24725402).

Answer: We added the reference to the manuscript (lines 92-93).

Line 122-129: Where all three foxes euthanized on the day they were found in the wild? If these foxes were first placed in a rehabilitation center please add how long the foxes have been kept, and the progression of the disease in time.

Answer: Yes, the foxes were euthanized shortly after their admission in the wildlife rehabilitation centers.

Line 141-160: Please describe consistently the histopathological changes at the different anatomical locations of the respiratory tract of the individual animals and whether these were associated with virus antigen. For example, virus antigen was observed in the olfactory mucosa of 1 fox, but it is unclear if histological lesions or virus antigen were observed in the olfactory mucosa of other foxes, or whether the olfactory mucosa was not studied for the other foxes.

Answer: We rephrased line 148-158 to describe in more detail the virus protein expression in relation to the histopathological changes in the respiratory tract. We also indicate whether these changes were studied in all foxes or not.

In addition, there seem to be some discrepancies in table S1. For Fox-Dorst nasal concha were negative for virus antigen, but histopathologic changes were not analyzed. Same for trachea of Fox-Heemskerk. I assume if the tissues are collected both analyses can be performed.

Answer: We thank the reviewer for highlighting the discrepancies in table S1. For fox-Dorst we did not sample nasal conchae. This was indicated in the text (line 152-153) and we changed accordingly the first row of Table S1. From fox-Heemskerk we analyzed the HE stain of the trachea. This information was also added to Table S1.

Line 155: Why do the authors refer to the histological changes of polioencephalitis? What are the differences compared to other viral encephalitis.

Answer: The histologic changes associated with viral protein were mainly located in the grey matter from which the neuropil is a part. The appropriate term for encephalitis in gray matter is polioencephalitis, which is similar to a viral encephalitis as described in this manuscript. We understand that this term can be confusing and for this we changed polioencephalitis to encephalitis (line 159 and Table S1), which is a more commonly accepted term and is also a correct description of the morphologic brain changes.

Line 174-175: it is unclear for which mutations was screened, please include.

Answer: The fox viruses were analyzed using FluServer (<https://fluserver.bii.a-star.edu.sg/>) for mutations increasing the zoonotic potential of the virus (lines 366-367). This analysis did not identify specific mutations in the fox viruses (lines 186-188).

Line 196: virus titers already peak at 48 hours post infection in mammalian cells.

Answer: The PB2-E627K mutation appears to increase virus replication from 48h post infection onwards on the mammalian cells, but not on the avian cells at 37°C. We modified the manuscript to clarify this, in lines 200 to 201.

Line 198-210: It is unclear what is tested for Table S3, it seems that for only one cell line this has been tested for the different time points. And why could this method not be used for the DF-1 cells?

Answer: Sample size did not allow us to analyze three way interactions with time, virus variant and cell type. Therefore, only interactions between time and virus or interactions between time and temperature could be analyzed for significant changes. Table S3 shows if there is a significant difference between the two virus variants at each time point and temperature. This table is for all cell lines.

Line 223-235: It is unclear if the virus passaging was performed multiple times? If this has only been performed once, the only conclusion that can be drawn from it is that E672K has an increased fitness in all cell lines.

Answer: Yes, this was also our conclusion in line 231-233. Although, the passaging experiment was only performed once, the results are confirmed by the experiment shown in Figure 4. This showed increased

replication for PB2-E627K on A549 and MDCK but not DF-1 cells at 37°C and increased replication for PB2-E627K on all cell lines at 33°C. This is in agreement with the faster selection of the PB2-627K variant at 33°C compared to 37°C and incomplete selection of the PB2-627K variant on DF-1 cells at 37°C. We modified the manuscript in lines 235 to 237 to clarify this conclusion is made both from the results in Figure 4 and the results from the passaging experiment

Line 247-250. As there is no information on when these foxes got infected it might be that virus replication in the respiratory tract has already peaked, and is therefore lower in the brain. As previous studies in foxes have shown, virus replication peaks early after exposure (PMID: 19046504). In ferrets similar kinetics are observed.

Answer: This is discussed in lines 265-268, and we now emphasize that in "earlier stages" of the infection replication may have occurred in the respiratory tract. Previous studies (including PMID:19046504) describe virus protein expression in the brain combined with positive staining in the lungs. Furthermore, the route of infection is of importance for tissue tropism as shown by PMID: 19046504 which only showed polio encephalitis after intravenous inoculation of foxes and not after feeding on infected bird carcasses which is most likely the route of infection of the wild foxes in our study.

Line 264-265: As the authors already mention before, it is unclear what happened in earlier stages of the infections, so you cannot conclude this virus is exclusive neurotropic. It is very likely that virus replication in the respiratory tract peaked early in the infection and is therefore not detected in the respiratory tract any more.

Answer: We indeed do not know if the virus is exclusive neurotropic, we only know that in the euthanized foxes the virus was found exclusively in the brain. We rephrased the sentence to reflect this, see lines 268 to 270.

Line 270-272: This is an overstatement.

Answer: Previous cases of H5 HPAI in mammals usually represent as a respiratory disease with high viral titers in the lung, and sometimes in the brain. Strong viral protein expression in the brain without detection in the lungs is now observed for the first time in wild foxes. We changed the sentence to exactly describe this (lines 274-276).

Line 283-296: The *in vivo* experiments show that both virus replicate very efficient in all cell lines at both temperatures. It might be that in some cells the viruses with E726K replicate somewhat faster, but this observation is not solid enough to extrapolate these finding to the *in vivo* situation in foxes (a quick google search showed that the body temp of foxes is 38.7 degrees Celsius).

Answer: The temperatures for this *in vitro* replication experiment were not intended to be identical to the internal body temperature of foxes but rather represent the approximate temperature of the mammalian and avian respiratory tract.

Line 308: How does an increased neurotropism contributes to the infection of wild red foxes?

Answer: The increased affinity of the virus for neurons may have allowed the virus to enter the brain using the olfactory bulb. This may have allowed infection of the foxes without replication of the virus in the respiratory tract or other organs.

Line 308-310: The fact that at the end-stage of disease virus is not detected in the respiratory tract, or the absence of evidence of virus spread among these 3 individuals, does not mean that this virus is not transmitted among foxes. For such conclusions other studies have to be performed (active surveillance, serology etc).

Answer: We agree. These lines refer to the three foxes analyzed in this study, where we found no evidence for shedding and no evidence for transmission based on phylogenetic analysis of the fox viruses. However spread between mammals should be closely monitored. We therefore added in lines

334 to 335: "surveillance for HPAI viruses in wild mammals should be expanded to closely monitor potential transmission between mammals".

December 1, 2022

Dr. Nancy Beerens
Wageningen Bioveterinary Research
Houtribweg 39
Lelystad
Netherlands

Re: Spectrum02867-22R1 (Highly pathogenic avian influenza H5N1 virus infections in wild red foxes (*Vulpes vulpes*) show neurotropism and adaptive virus mutations)

Dear Dr. Nancy Beerens:

Thank you for submitting your manuscript to Microbiology Spectrum. Your revised manuscript was sent back to Reviewer 2 to assess whether they would consider your modifications sufficient to advise a publication in Microbiology Spectrum. While they were overall satisfied with your modifications, they still recommend to tune down your conclusions regarding the transmissibility and neurotropism of the studied viruses in foxes. I invite you to address the remaining comments of Reviewer 2, please see below.

Link Not Available

Sincerely,

Mathilde Richard

Journals Department
Reviewer comments:

Reviewer #2 (Comments for the Author):

The authors have addressed the majority of my point, however, I still have some concerns regarding the following:

The statement in the abstract 'limited virus shedding was detected suggesting the virus was not transmitted between the foxes.':

It is not uncommon to see very limited virus antigen (or associated lesions) in the lower respiratory tract of H5N1 infected mammals at later stages of the infection. In foxes consumption of H5N1-virus infected carcasses did not result in the detection of virus antigen at 7 days post inoculation (PMID: 19046504). Another example, virus antigen was only detected in 1 out of 3 H5N1 virus inoculated ferrets at 7 days post inoculation (PMID 22278228). There is thus a large heterogeneity with respect to replication in the respiratory tract and the duration of virus replication in the respiratory tract (and in experimental setting this often depends on route of inoculation). The authors do not know the course of the infection in the respiratory tract in these foxes. In addition, even if this was known, it would be difficult to extrapolate this to a transmission event. In my view speculations concerning the transmissibility of this virus among foxes must be adjusted throughout the manuscript.

Another concern is about the claim that this virus has an increased neurotropism. Only 3 foxes were studied and it is unclear how representative these are for all H5N1 virus infected foxes. Maybe many other foxes got respiratory disease and survived. Furthermore, at later stages of the infection it is not uncommon to find virus replication in the CNS, without evidence for virus antigen in the respiratory tract. This has been studied extensively in ferrets for H5N1 viruses (PMID: 2138913; PMID: 27448390; PMID 22278228). In order to make the claim that this virus has an increased neurotropism (or more neuroinvasive) this virus should be studied side-by-side with other H5N1 viruses in a controlled setting. In my opinion this claim needs to be adjusted throughout the manuscript, for example in line 274-277: 'Thus, the lack of virus replication in the respiratory system of the wild foxes is interesting, and the detection in the brain suggest the current HPAI H5N1 2021-2022 viruses have an increased neurotropism.'

Staff Comments:

Preparing Revision Guidelines

Please return the manuscript within 60 days; if you cannot complete the modification within this time period, please contact me. If you do not wish to modify the manuscript and prefer to submit it to another journal, please notify me of your decision immediately so that the manuscript may be formally withdrawn from consideration by Microbiology Spectrum.

Reviewer #2 (Comments for the Author):

The authors have addressed the majority of my point, however, I still have some concerns regarding the following:

The statement in the abstract 'limited virus shedding was detected suggesting the virus was not transmitted between the foxes.':

It is not uncommon to see very limited virus antigen (or associated lesions) in the lower respiratory tract of H5N1 infected mammals at later stages of the infection. In foxes consumption of H5N1-virus infected carcasses did not result in the detection of virus antigen at 7 days post inoculation (PMID: 19046504). Another example, virus antigen was only detected in 1 out of 3 H5N1 virus inoculated ferrets at 7 days post inoculation (PMID 22278228). There is thus a large heterogeneity with respect to replication in the respiratory tract and the duration of virus replication in the respiratory tract (and in experimental setting this often depends on route of inoculation). The authors do not know the course of the infection in the respiratory tract in these foxes. In addition, even if this was known, it would be difficult to extrapolate this to a transmission event. In my view speculations concerning the transmissibility of this virus among foxes must be adjusted throughout the manuscript.

Answer:

We are sorry to read the reviewer still has concerns regarding the possibility of virus transmission between mammals. The phylogenetic analysis performed in our study provides evidence for independent transmission, as the mammalian viruses are not clustering and are not highly related (Figure 3, text lines 162-172). In addition, airborne transmission of avian influenza viruses has been shown to require several additional mutations (PMID: 22723413 , line 83-93), that were not detected in our study. We hypothesized that the risk on virus transmission may be low since limited virus shedding was observed for the three foxes. Indeed, as this was the end-stage of infection, and just three animals, we cannot exclude that earlier in the infection cycle virus shedding did occur. This was already explicitly mentioned in lines 259-262, but we now made several additional changes to avoid misunderstanding on this point (lines 26-29, 309-310).

Another concern is about the claim that this virus has an increased neurotropism. Only 3 foxes were studied and it is unclear how representative these are for all H5N1 virus infected foxes. Maybe many other foxes got respiratory disease and survived. Furthermore, at later stages of the infection it is not uncommon to find virus replication in the CNS, without evidence for virus antigen in the respiratory tract. This has been studied extensively in ferrets for H5N1 viruses (PMID: 2138913; PMID: 27448390; PMID 22278228). In order to make the claim that this virus has an increased neurotropism (or more neuroinvasive) this virus should be studied side-by-side with other H5N1 viruses in a controlled setting. In my opinion this claim needs to be adjusted throughout the manuscript, for example in line 274-277: 'Thus, the lack of virus replication in the respiratory system of the wild foxes is interesting, and the detection in the brain suggest the current HPAI H5N1 2021-2022 viruses have an increased neurotropism.'

Answer:

As we indeed did not compare this virus to previous viruses in this study, we removed the word "increased". We now state that "strong neurotropism" was observed for the HPAI H5N1 2020-21 viruses in the manuscript. This change was made in lines 270, 306-307.

December 23, 2022

Dr. Nancy Beerens
Wageningen Bioveterinary Research
Houtribweg 39
Lelystad
Netherlands

Re: Spectrum02867-22R2 (Highly pathogenic avian influenza H5N1 virus infections in wild red foxes (*Vulpes vulpes*) show neurotropism and adaptive virus mutations)

Dear Dr. Nancy Beerens:

Your manuscript has been accepted, and I am forwarding it to the ASM Journals Department for publication. You will be notified when your proofs are ready to be viewed.

Sincerely,

Mathilde Richard
Editor, Microbiology Spectrum
